# SPoT: Subpixel Placement of Tokens in Vision Transformers

Martine Hjelkrem-Tan[1], Marius Aasan[1], Gabriel Y. Arteaga[1], and Adín Ramírez Rivera[1]

[1] Department of Informatics, University of Oslo

`{matan, mariuaas, gabrieya, adinr}@uio.no`

**Reviewed on OpenReview:** <https://openreview.net/forum?id=XrBzSmzAVo>

## Abstract

Vision Transformers naturally accommodate sparsity, yet standard tokenization methods confine features to discrete patch grids. This constraint prevents models from fully exploiting sparse regimes, forcing awkward compromises. We propose Subpixel Placement of Tokens (SPoT), a novel tokenization strategy that positions tokens continuously within images, effectively sidestepping grid-based limitations. With our proposed oracle-guided search, we uncover substantial performance gains achievable with ideal subpixel token positioning, drastically reducing the number of tokens necessary for accurate predictions during inference. SPoT provides a new direction for flexible, efficient, and interpretable ViT architectures, redefining sparsity as a strategic advantage rather than an imposed limitation.

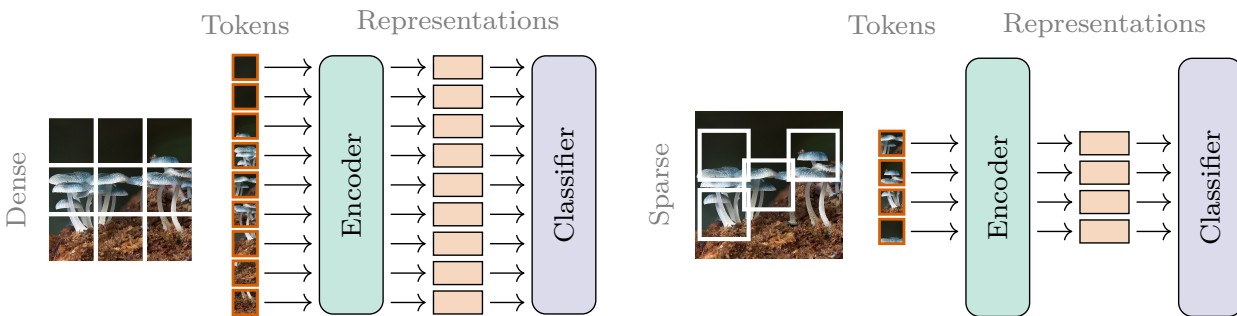

**Figure 1:** (Left) A standard ViT splits the image into a fixed grid of non-overlapping patches. (Right) With SPoT, a continuously sampled set of subpixel-precise patches are extracted.

## 1 Introduction

Sparsity—the fine art of doing more with less—is an attractive prospect in systems design and modeling. As models grow ever larger, sparse features alleviates the computational demands of a model to provide lower latency, lower memory overhead, and higher throughput—all important properties for real-time applications. Incidentally, sparse selection of features offers inherent interpretability and transparency for increasingly complex models (Tibshirani, 1996). Clever adaptations of the Vision Transformer (ViT) (Dosovitskiy et al., 2021) architecture have shown that this family of models can handle sparse inputs remarkably well (Liu et al., 2023; Chen et al., 2023; Rao et al., 2021; Yin et al., 2022; Bolya et al., 2023), accelerating inference by selectively processing a reduced subset of the input. Sparsification has even indirectly inspired entirely new paradigms for efficient unsupervised training in the form of masked image modeling (MIM) (He et al., 2022; Zhou et al., 2022; Oquab et al., 2024).

---

Code available at: <https://github.com/dsb-ifi/SPoT>

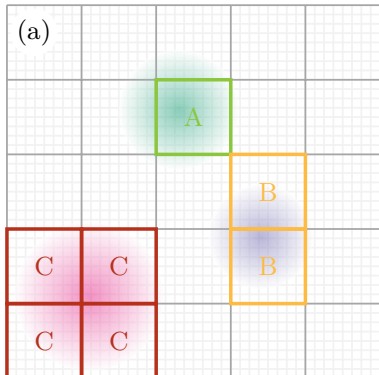 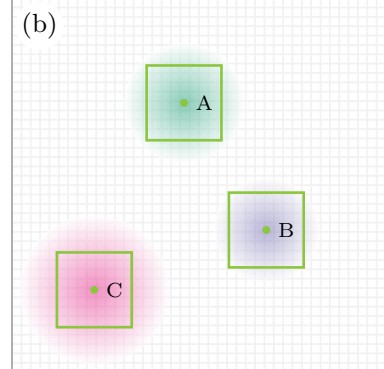

**Figure 2: Grids cannot align over all key features.** (a) A $5 \times 5$ patch grid (gray) with three optimal region placements for sparse feature selection. The **green** patch is well aligned (A), **yellow** straddles two cells (B), and **red** lies on a corner (C) and leaks into *four* cells. Translating the grid only swaps which peak is misaligned—one patch is always bad. (b) Our subpixel tokenizer drops fixed-size windows (**green** squares) directly on each peak, eliminating the alignment trade-off while still allowing conventional grid tokens when they *are* well aligned.

However, in carefully studying the fine print of Dosovitskiy et al.'s (2021) work, one notes an insistence on aligning features with an underlying grid, mirroring the structure inherited by its language counter-part (Vaswani et al., 2017; Devlin et al., 2019) where inputs are naturally represented as sequences of discrete tokens. This discretization might seem natural—after all, are not pixels fundamentally discrete? Our hypothesis is that this adherence turns sparsity into an awkward dance; forcing the selection of entire tiles, even if the true optimal feature set hides in-between rigid lines. Like eating soup with a fork: possible, but decidedly inefficient and frustrating.

We propose a simple remedy with *Subpixel Placement of Tokens* (SPoT). By allowing patches to occupy continuous subpixel positions instead of constraining features to a discrete grid, we expand our modeling toolkit to include gradient based search and sampling for discovering optimal sparse feature sets. Figure 1 succinctly illustrates the core idea, while the example in Figure 2 shows the limitations of a discrete grid-based approach in reducing tokens while preserving predictive quality. Our contributions include:

- We propose *SPoT*, a novel tokenization framework placing features at continuous subpixel positions, significantly enhancing the robustness and efficiency of ViTs.

- We introduce *Oracle-guided Neighborhood search* (SPoT-ON), an analysis tool to empirically quantify optimal subpixel positions, showing that carefully selected sparse placements can outperform dense grids with only $\sim 12.5\%$ of the original tokens. SPoT-ON provides an empirical upper bound on performance that can be gained by only changing what the model sees, and we show that optimal regions discovered with one model improve performance in another.

- We systematically investigate *spatial priors for subpixel token placement*, and find that dense regimes prefer coverage, while sparse regimes benefit from center bias and saliency-driven priors.

## 2 Sparse Visual Bag-of-Words and ViTs

At first glance it might seem like a ViT is designed to process an image globally via partitioning an image into patches. However, there is nothing in the transformer architecture that requires a discrete partition. Because self-attention is permutation-invariant, a ViT encoder effectively treats its input tokens as an unordered multiset; a visual bag-of-words, analogous to BERT (Devlin et al., 2019). This observation suggests we need not restrict tokens to a grid, leading to our formulation of sparse feature selection for ViTs.

We denote a ViT encoder as $g_\theta \colon \mathcal{I} \times \Omega \to \mathbb{R}^d$, where $\mathcal{I}$ is a dataset of source images, and $\Omega$ is a space of positions from which to sample image features. For example, with standard tokenization $\Omega_{\text{grid}}$ is a fixed,

discrete set of non-overlapping square patches tiling the image with a fixed window size on a grid of pixels. The sparse feature selection (SFS) problem can then be formulated as

$$\min_{\phi} \mathbb{E}_{S \sim p_\phi}\big[\mathcal{L}(g_\theta(I, S), y)\big] \text{ s.t. } S \subseteq \Omega, \ |S| \ll |\Omega|. \tag{1}$$

In other words, for each image $I$ we are looking for a probability distribution $p_\phi$ over subsets of $\Omega$ that minimizes a loss function $\mathcal{L}$[1]. We note that for the discrete non-overlapping case of $\Omega_{\text{grid}}$, there is an implicit assumption that sampling of $S$ is done without replacement, since sampling the same feature more than once is unlikely to improve model performance. Three specific issues arise from the ViT sparse sampling problem:

1. **Interdependence**: Transformers process tokens as a set. This means that the marginal distribution of one token is dependent on the inclusion of other tokens. Furthermore, the optimal distribution $p_\phi$ for a given image may vary depending on the choice of number of tokens.

2. **Combinatorial search**: The discrete nature of $\Omega_{\text{grid}}$ means that selecting a subset of tokens is combinatorial knapsack problem. This makes search difficult and gradient methods intractable, particularly since cardinality-constrained subset selection is NP-hard (Nemhauser & Wolsey, 1978).

3. **Misalignment**: By quantizing patches to a fixed grid, key patterns for discriminating an image could be missed in SFS. Concretely, if the grid imposed by $\Omega_{\text{grid}}$ is misaligned with key features in the image, SFS could be challenging, as a central shape or texture may be spread over multiple patches, making subset selection more challenging.

These issues hinder efficient optimization of SFS under standard tokenization.

## 3   Subpixel Placement of Tokens: SPoT

We propose a more flexible tokenization scheme to tackle SFS problems in ViTs. Instead of considering $\Omega$ as a fixed discrete partition, we instead imagine $\Omega_{\text{subpix}} = [0, H - 1] \times [0, W - 1]$ as a continuous space of subpixel positions from which to select features within a $H \times W$ image. Put simply, we parametrize a subset of positions $S = \{s_1, \ldots, s_m\}$ as a set of points of interest from which to extract features from within an image. By sampling tokens from continuous subpixel positions, our tokenizer directly addresses the intrinsic *misalignment* issue imposed by traditional grid-based methods, as illustrated in Figure 2. To tackle the *combinatorial search* problem, we use a bilinear interpolation function $q$ and window size $k$, such that each subpixel position $s_i = (h, w)$ provides an extracted feature

$$I_q(s_i; k) = I_q(h - \tfrac{k}{2} : h + \tfrac{k}{2}, \ w - \tfrac{k}{2} : w + \tfrac{k}{2}). \tag{2}$$

This allows us to formulate SFS as a continuous, probabilistic optimization problem rather than an intractable discrete subset-selection. The key insight is that our novel tokenizer allows us to (1) investigate placing *different priors* on $p_\phi$, and (2) use *gradient based optimization* to search for $S$ by way of gradients through $I_q$. Since we select $q$ to be bilinear, its partial derivatives w.r.t. $s$ exist everywhere except at pixel boundaries, so gradients propagate cleanly back to the placements $\{s_1, \ldots, s_m\}$. We note that subpixel tokens do not impose any constraint on non-overlapping patches.

Since $\Omega_{\text{grid}} \subseteq \Omega_{\text{subpix}}$, patch tokenization is just a special case of our tokenization method. This means that models can be evaluated with the exact same features as a standard patch-based ViT by letting $S = \Omega_{\text{grid}}$. Our tokenizer extends existing work showing that more generalized tokenizers can be constructed to be modularly commensurable to standard ViT models, and we adopt their kernelized positional embedding (Aasan et al., 2024).

### 3.1   Spatial Priors

By allowing subpixel freedom in token placement, we lose the implicit spatial prior that discrete grids naturally encode. Hence the shift to a continuous domain raises a dilemma: in sparse regimes, what should

---

[1]Typically standard cross-entropy is used for $\mathcal{L}$.

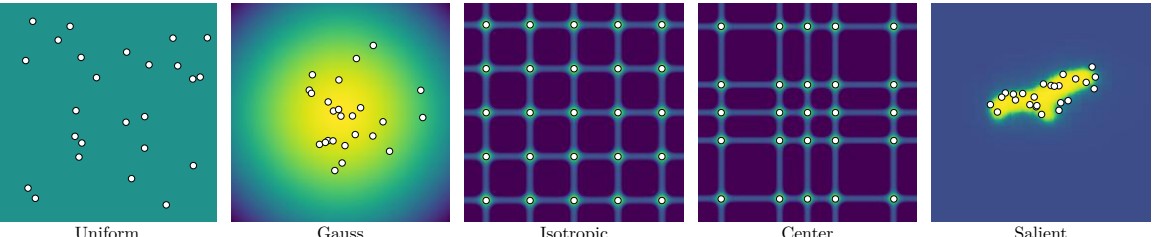

**Figure 3:** Different sampling priors which can be employed with SPoT. The Sobol prior (not figured) produces uniform quasirandom placements with explicit constraints on coverage.

guide our choice of token positions? An appropriately selected prior enables efficient sparse representations while preserving performance, whereas an ill-suited one may lead to substantial degradation. We compare several spatial priors, each encoding different assumptions about feature importance and spatial distribution, illustrated in Figure 3:

- **Uniform**: randomly samples locations with no spatial bias, assuming all regions are equally important.
- **Gaussian**: randomly samples locations with a central bias, which encodes a prior belief that subjects are typically centered in images.
- **Sobol**: provides quasirandom sampling aimed at uniform coverage while reducing overlap (Sobol, 1967).
- **Isotropic**: deterministically distributes tokens evenly in a subpixel grid, emphasizing coverage.
- **Center**: deterministically distributes tokens evenly with slight central-bias, commonly seen in classification datasets (Szabó & Horváth, 2022).
- **Salient**: encodes object-centric bias by placing tokens based on regions identified as visually salient from a pretrained saliency model (Lüddecke & Ecker, 2022).

Moreover, we highlight that a prior could be learned to suit a particular problem. However, we note that interdependency requires a more complex parametric family than a univariate spatial distribution. We therefore leave this step as future work, and focus on the proposal of the continuous positions of tokens and its evaluation.

### 3.2 Oracle Neighborhoods: SPoT-ON

In addition to investigating spatial priors, we also look to directly explore differentiable optimization for token placement. To probe for ideal choices of $S = \{s_1, \ldots, s_m\}$, we optimize a constrained version of the SFS problem given by equation 1 directly for each image. We freeze the encoder $g_\theta$, and directly apply gradient search per image $I$ to optimize

$$\arg\min_S \left[ \mathcal{L}(g_\theta(I, S), y) \right] \text{ s.t. } S \subseteq \Omega_{\text{subpix}}, \ |S| = m \tag{3}$$

for a set number of tokens $m$ with initial positions $S^0 \sim p_\phi$ sampled from a chosen prior $p_\phi$. This provides an *Oracle Neighborhood* (ON) adjustments of the initial placements for SPoT. SPoT-ON reveals ideal locations for classifying each image, which allows us to ascertain the existence of an optimal set of positions $S$ for each image, and estimate an upper bound on performance gain from effective token sampling. We specify that SPoT-ON incurs a higher computational cost for classification, and is *not intended as a practical solution for inference*. Rather, it acts as a tool for analyzing the nature of sparse ViTs, demonstrating the potential of learnable token positions.

## 4 Experiments: Case Studies

We examine SPoT by adapting two standard ViT models (Steiner et al., 2022), trained on ImageNet-21k and ImageNet-1k (CLS-IN21k, CLS-IN1k) and a self-supervised Masked Autoencoder (MAE-IN1k) (He et al.,

**Table 1:** Oracle accuracy of grid-constrained and off-grid patch representations in extreme sparse setting with 12.5% of tokens. The grid-based configuration mimics the discrete patch selection of standard ViTs. The off-grid configuration permits subpixel placement in continuous space. Results demonstrate that allowing continuous positioning enhances representational quality under sparse token regimes. We report the performance of initially sampled points (SPoT) and oracle optimized placements (SPoT-ON).

| Method | lr | Steps | Acc@1 (%) | Oracle |
|--------|-----|-------|-----------|--------|
| SPoT | – | – | 61.7 | |
| **Grid** | | | | |
| SPoT-ON | 3e-3 | 5 | 66.2 | ✓ |
| SPoT-ON | 1e-2 | 10 | 74.0 | ✓ |
| **Subpixel** | | | | |
| SPoT-ON | 3e-3 | 5 | **90.2** | ✓ |
| SPoT-ON | 1e-2 | 10 | **90.9** | ✓ |

**Table 2:** Accuracy (%) for different spatial initialization priors in extreme sparse setting with 25 tokens. We show SPoT performance (Acc@1) and the potential increase in performance obtained under oracle optimization using SPoT-ON (Oracle $\Delta$).

| Prior | k-NN (%) | Acc@1 (%) | Oracle $\Delta$ |
|-------|----------|-----------|-----------------|
| **SPoT CLS-IN21k** | | | |
| Uniform | 45.23±0.10 | 44.05 | ↑34.22 |
| Gaussian | 45.27±0.10 | 45.22 | ↑32.83 |
| Sobol | 46.48±0.20 | 43.67 | ↑35.83 |
| Isotropic | 48.19 | 46.85 | ↑34.85 |
| Center | 52.18 | 52.45 | ↑31.07 |
| Salient | 56.83±0.26 | 55.71 | ↑31.90 |
| **SPoT MAE-IN1k** | | | |
| Uniform | 49.72±0.13 | 56.71 | ↑31.44 |
| Gaussian | 49.49±0.33 | 57.58 | ↑29.63 |
| Sobol | 53.54±0.39 | 60.62 | ↑29.40 |
| Isotropic | 54.56 | 61.72 | ↑29.21 |
| Center | 55.61 | 62.83 | ↑26.70 |
| Salient | 60.80±0.08 | 66.13 | ↑26.59 |

2022). All models utilize the ViT-B/16 architecture (Dosovitskiy et al., 2021). Supervised models initialize weights from TIMM (Wightman, 2019), while MAE uses official *pre-trained* weights. To integrate our subpixel tokenizer, each model undergoes a 50-epoch *retrofitting* step on ImageNet-1k (Deng et al., 2009), after which the MAE-based model is further *fine-tuned* for classification over 100 epochs, aligning with its original protocol (He et al., 2022). Additional details are given in Section 5.2. We explore the effectiveness and properties of our approach through four targeted case studies addressing grid limitations, object-centric priors, oracle guidance preferences, and transferability of guided placements.

### 4.1 Are Grids an Inherent Limitation of ViTs?

In our first case study, we investigate whether moving away from fixed-grid token representations toward subpixel placements in continuous space enhances representational quality. Traditional grid-based representations restrict tokens to fixed intervals, which often require multiple patches to cover important features adequately, as illustrated in Figure 2. Subpixel placement, on the other hand, allows tokens to align precisely with these features, potentially enabling more efficient representations with a sparse set of tokens.

To investigate grid versus off-grid representations, we design an experiment using SPoT-ON to directly compare continuous subpixel placement with discrete, grid-based positioning, all under a fixed token budget of 12.5% of the standard 196 in ViT-B/16 architectures. For the discrete setting, the learned subpixel positions were mapped to their nearest locations on a standard $14 \times 14$ token grid, mimicking a conventional ViT configuration. We consider two optimization configurations: one with a learning rate of $3 \times 10^{-3}$ over 5 optimization steps, and another with a higher learning rate of $1 \times 10^{-2}$ over 10 steps.

The results in Table 1 clearly demonstrate the advantage of subpixel placement, which achieves at least a 16.9 percentage point improvement in accuracy over the grid-constrained method. Interestingly, increasing both the learning rate and the number of optimization steps allows the grid-based approach to discover more effective token positions. Nevertheless, the constraints of discrete, grid-based positioning hinders performance, even under more aggressive optimization. The consistent performance gains highlights significant benefits of continuous, subpixel token placement in resource constrained settings.

**Finding 1:** Off-grid token placement enables greater flexibility and yields substantially better performance than grid-based approaches under sparse token settings.

## 4.2 Do Object-Centric Priors Improve Predictions?

To investigate spatial priors and how they interact with oracle supervision in the sparse setting, we initialize our adaptive sampler using the spatial priors introduced Section 3.1. Each initialization defines the starting coordinates $S^0$ of token placements, which are subsequently refined by SPoT-ON to minimize classification loss with a learning rate of $3 \times 10^{-3}$ over 5 optimization steps.

Table 2 reports the resulting downstream accuracy for both supervised and self-supervised backbones in the sparse setting with a token budget of 25 tokens (12.5% of original). We observe that sampling from saliency heatmaps yields the highest performance both out-of-the-box and after oracle supervision. This aligns with common intuition, object-centric features are more relevant to the classification task. The center grid prior also shows higher performance, very likely due to the center bias in ImageNet images. Finally, grid-based initializations (e.g., regular grid, center grid, and Sobol) consistently lead to higher accuracy than the uniform random and Gaussian-stochastic priors, as these result in overlapping placements that are less efficient than structured alternatives.

Comparing these results to Table 5, we see that the benefit of object centric sampling disappears for higher token budgets. Instead, the best performing prior is the regular grid, which ensures broad spatial coverage rather than concentrating solely on the object. This reveals a surprising inductive bias—coverage is more critical than object-centricity for classification under high token budgets. We hypothesize that this is because the information provided by object-focused features quickly saturates, and with higher token budgets the model benefits from the broader context provided by even coverage of the image.

> **Finding 2:** Object-centric priors yield higher performance in sparse regimes. In dense regimes, even and structured coverage provides better performance.

## 4.3 Does Oracle Guidance Prefer Salient Regions?

Intuitively, object-centric priors should help a classifier, but do they actually steer our oracle-guided tokenizer? We investigate whether oracle gradient search in SPoT-ON moves tokens towards pixels with higher class saliency.

We design an experiment as follows; starting with a isotropic prior, we optimize trajectories $s^{(0)}, \dots, s^{(t)}$ via oracle gradient search following equation 3. Our goal is to measure the shift of token placements toward higher-saliency regions between the initial position $s^{(0)}$ and the final position $s^{(t)}$. Given a saliency mask $M$, we compute a score for placements $s$ by

$$\text{score}(s) = \frac{1}{k^2} \sum_{i=1}^{k^2} M_q(s; k)_i. \tag{4}$$

The *relative saliency gain* for each trajectory is given by

$$\text{RSG} = \frac{\text{score}(s^{(t)}) - \text{score}(s^{(0)})}{\text{score}(s^{(0)})}. \tag{5}$$

Table 3 shows the result of averaging relative saliency gains over ImageNet1k, showing that there is a slight gain in saliency scores for each of our three models. However, the results are not significant enough to claim that saliency alone guides the placements during the oracle search.

We illustrate thirty examples with different trajectories in Fig. 4, which sheds more light on the behavior of SPoT-ON. While trajectories are drawn to discriminative regions, such as spots on a ladybug (left) or hands of a clock (center right), other placements seem more arbitrary, and even loop back on themselves. The oracle often positions tokens close to—rather than on—the object; providing context that self-attention can exploit. Hence, *interdependency* rather than saliency alone, drives the final placements.

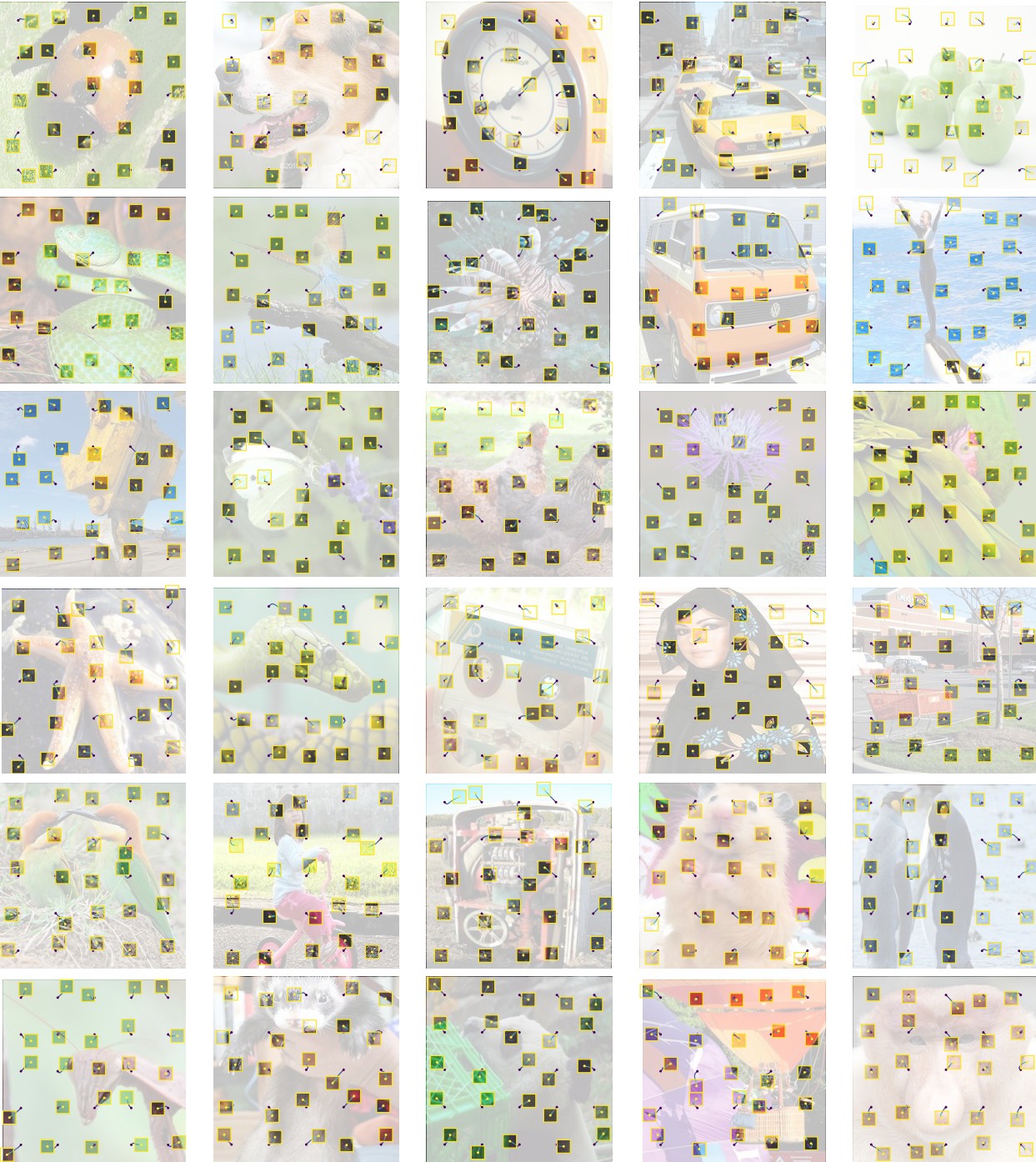

**Figure 4:** Illustration of oracle placements with 25 tokens with SPoT-ON. By optimizing our oracle-neighborhood search equation 3 all the way through the model, the oracle discovers optimal placement of points, yielding an accuracy of 90.9% on ImageNet1k with only $\sim$ 12.5% of the tokens. Trajectories are colored starting with dark purple for initial points, with endpoints colored bright yellow.

**Finding 3:** While oracle gradient search yields a slight bias toward higher-saliency pixels, the results are not highly significant. The results suggest that token interdependency — as opposed to pure object saliency — is a predominant factor for optimal placements.

**Table 3:** Quantitative analysis on object-seeking behavior with SPoT-ON. For each region, we compute the average saliency in a $16 \times 16$ window centered on each point. We compare the saliency scores of initial placements compared to oracle placements, and compute the relative saliency gain (RSG) over ImageNet1k.

| | RSG (%) | | | |
|---|---|---|---|---|
| **Backbone** | **25 Tok.** | **49 Tok.** | **100 Tok.** | **196 Tok.** |
| CLS-IN21k | ↑0.34 | ↑0.33 | ↑0.47 | ↓0.14 |
| CLS-IN1k | ↑0.77 | ↑0.77 | ↑0.75 | ↓1.53 |
| MAE-IN1k | ↑0.49 | ↑0.54 | ↑0.51 | ↓0.08 |

**Table 4:** Transfer properties of SPoT-ON positions between models, in the sparse setting with a 12.5% token budget. We optimize oracle positions using gradient optimization for a source model, initialized with the isotropic prior. We then evaluate the discovered points in an independently trained target model. Each model sees a significant increase in performance, even from points derived from an independent model.

| | Acc@1 (%) (25 Tokens) | | |
|---|---|---|---|
| **Source → Target** | **Original** | **Transfer** | **Δ** |
| CLS-IN1k → CLS-IN21k | 46.85 | 54.06 | ↑7.21 |
| MAE-IN1k → CLS-IN21k | 46.85 | 56.71 | ↑9.86 |
| CLS-IN21k → CLS-IN1k | 33.52 | 37.91 | ↑4.39 |
| MAE-IN1k → CLS-IN1k | 33.52 | 39.19 | ↑5.67 |
| CLS-IN21k → MAE-IN1k | 61.72 | 68.50 | ↑6.78 |
| CLS-IN1k → MAE-IN1k | 61.72 | 67.81 | ↑6.09 |

## 4.4 Do Oracle Guided Placements Transfer?

If discovered token placements with oracle guidance captures structure rather than model-specific quirks, *a placement learned by one model should benefit another*. To test this, we investigate *transferability* of token placements. Given two independently trained models, $g_A, g_B$, we independently optimize two feature sets; $S_A, S_B$, respectively, following equation 3. Both feature sets are initialized with the same isotropic spatial prior $S_0$. Then, for target labels $y$, we compute the difference in accuracy from the initial placements $S_0$ to the optimized placements with the alternate model, i.e.,

$$\Delta = \mathbb{E}[g_A(I, S_0) = y] - \mathbb{E}[g_A(I, S_B) = y], \tag{6}$$

with symmetrical computations for model $g_B$ on $S_A$. The results can be found in Table 4. Our experiments show that placements discovered with one model transfers to yield improvement in performance in a different, independently trained model.

> **Finding 4:** Discovered positions via SPoT-ON generalize between models; a set of placements optimized via one model will improve results with another independently trained model in sparse regimes.

## 5 Extended Experimental Results

We present the performance of SPoT under varying sparsity configurations and compare it against baseline models, including the supervised backbones from TIMM (Wightman, 2019) and the officially *fine-tuned* MAE model (He et al., 2022). For clarity, all baselines are denoted as ViT-B/16 in Table 5. To evaluate the baselines under sparsity constraints, we apply PatchDropout (Liu et al., 2023), which randomly drops input patches during inference.

The results in Table 5 reveal several noteworthy observations. First, the self-supervised MAE model (He et al., 2022) consistently outperforms its supervised counterparts under sparse configurations. This advantage is likely due to its pre-training objective, which inherently involves reconstructing inputs from partial observations, thereby fostering robustness to patch dropout. Second, we observe that under full-token conditions, SPoT achieves marginally higher performance than both the supervised and self-supervised baselines.

**Table 5:** Classification top-1 and kNN accuracies for supervised and and self-supervised models using different token priors. We find that center-bias in spatial priors is beneficial in sparse regimes, while coverage becomes more important as token budgets increase.

| Model | Prior | Oracle | 25 Tokens Acc@1 | kNN | 49 Tokens Acc@1 | kNN | 100 Tokens Acc@1 | kNN | 196 Tokens Acc@1 | kNN |
|---|---|---|---|---|---|---|---|---|---|---|
| **CLS-IN21k** | | | | | | | | | | |
| ViT-B/16 | Patch Grid | | 24.72 | 27.86 | 56.29 | 57.19 | 78.75 | 78.77 | 85.11 | 83.96 |
| SPoT-B/16 | Uniform | | 44.05 | 45.23 | 67.77 | 66.38 | 79.64 | 78.03 | 83.76 | 81.85 |
| SPoT-B/16 | Gaussian | | 45.22 | 45.27 | 68.64 | 66.96 | 79.75 | 77.74 | 83.45 | 81.48 |
| SPoT-B/16 | Sobol | | 43.67 | 46.48 | 69.02 | 68.60 | 81.63 | 79.35 | 84.66 | 82.62 |
| SPoT-B/16 | Isotropic | | 46.85 | 48.19 | **70.61** | **70.29** | **82.20** | **80.73** | **85.15** | **83.42** |
| SPoT-B/16 | Center | | **52.45** | **52.18** | 69.22 | 68.16 | 80.84 | 78.56 | 84.01 | 82.23 |
| SPoT-B/16 | Salient | ✓ | 55.71 | 56.65 | 72.89 | 72.38 | 79.91 | 80.56 | 84.56 | 82.59 |
| SPoT-ON-B/16 | Isotropic | ✓ | 81.70 | 70.65 | 94.28 | 88.58 | 95.97 | 92.92 | 96.12 | 93.52 |
| **CLS-IN1k** | | | | | | | | | | |
| ViT-B/16 | Patch Grid | | 9.24 | 12.05 | 41.05 | 44.38 | 71.22 | 71.41 | 79.14 | 77.64 |
| SPoT-B/16 | Uniform | | 29.87 | 33.88 | 60.64 | 60.84 | 74.44 | 73.18 | 79.38 | 77.36 |
| SPoT-B/16 | Gaussian | | 29.27 | 33.07 | 60.47 | 60.23 | 74.37 | 72.82 | 79.02 | 77.00 |
| SPoT-B/16 | Sobol | | 30.67 | 35.23 | 64.42 | 63.88 | 76.45 | 75.18 | 79.96 | 78.17 |
| SPoT-B/16 | Isotropic | | 33.52 | 37.84 | **66.18** | **66.25** | **77.58** | **76.29** | **80.61** | **79.04** |
| SPoT-B/16 | Center | | **39.91** | **42.47** | 63.04 | 62.65 | 75.41 | 73.63 | 79.32 | 77.71 |
| SPoT-B/16 | Salient | ✓ | 39.83 | 43.72 | 66.32 | 66.00 | 74.36 | 75.25 | 79.54 | 78.03 |
| SPoT-ON-B/16 | Isotropic | ✓ | 73.99 | 74.42 | 94.21 | 90.11 | 95.79 | 93.61 | 96.04 | 93.97 |
| **MAE-IN1k** | | | | | | | | | | |
| ViT-B/16 | Patch Grid | | 55.43 | 48.85 | 70.69 | 67.15 | 79.53 | 78.41 | 83.60 | 82.07 |
| SPoT-B/16 | Uniform | | 56.71 | 49.72 | 73.22 | 65.85 | 80.53 | 74.76 | 82.78 | 78.21 |
| SPoT-B/16 | Gaussian | | 57.58 | 49.49 | 72.51 | 65.59 | 80.31 | 74.52 | 82.55 | 77.90 |
| SPoT-B/16 | Sobol | | 60.62 | 53.54 | 75.71 | 68.71 | 82.19 | 76.24 | 83.51 | 79.09 |
| SPoT-B/16 | Isotropic | | 61.72 | 54.56 | **76.84** | **70.02** | **82.76** | **77.24** | **83.89** | **79.53** |
| SPoT-B/16 | Center | | **62.83** | **55.61** | 74.63 | 67.31 | 81.06 | 75.20 | 82.97 | 78.54 |
| SPoT-B/16 | Salient | ✓ | 66.13 | 60.80 | 77.10 | 72.24 | 81.46 | 77.25 | 81.64 | 79.13 |
| SPoT-ON-B/16 | Isotropic | ✓ | 90.93 | 79.73 | 94.87 | 87.87 | 96.09 | 90.76 | 96.24 | 91.28 |

**Table 6:** Analysis on harmful spatial priors and adversarial oracles in sparse regimes. The background prior samples from inverse saliency maps; the boundary prior samples with image edge bias. Ascent shows accuracy under worst-case token placements, discovered via SPoT-ON. Label obfuscation optimizes placements for randomized labels. Each case is compared with baseline SPoT performance using the isotropic prior—the performance drop is shown to the right of each score.

| Backbone | Acc@1 (%) (25 Tokens) Backgrd. | Boundary | Ascent | Lab.Obf. |
|---|---|---|---|---|
| CLS-IN21k | 40.80↓ 6.85 | 10.68↓36.17 | 13.75↓33.09 | 1.17↓45.69 |
| CLS-IN1k | 20.06↓13.46 | 4.15 ↓29.37 | 5.90 ↓27.82 | 2.68↓33.52 |
| MAE-IN1k | 31.89↓29.83 | 10.83↓50.89 | 16.54↓45.18 | 1.11↓60.61 |

This suggests that even in dense settings, additional gains can be realized by leveraging the flexibility introduced by subpixel representations. Third, as the level of sparsity increases, SPoT consistently surpasses all baselines, regardless of spatial prior. Notably, performance is further improved with priors that promote spatial coverage, compared to stochastic uniform sampling, which demonstrates the importance of an appropriate token placement scheme under sparsity constraints.

In Figure 5 we show image throughput versus accuracy, comparing SPoT with the baselines across varying sparsity levels. As sparsity increases, throughput improves significantly, albeit with an associated trade-off in accuracy. Notably, SPoT achieves the most favorable trade-off, maintaining substantially more of the full-model accuracy while enabling higher throughput than competing approaches. Further, we observe only slight variation in throughput between the models at each sparsity level, indicating that SPoT incurs very minimal computational overhead compared to baselines. We also include our oracle-guided variant SPoT-ON in the figure, which illustrates a ceiling on achievable performance when placements are ideally sampled.

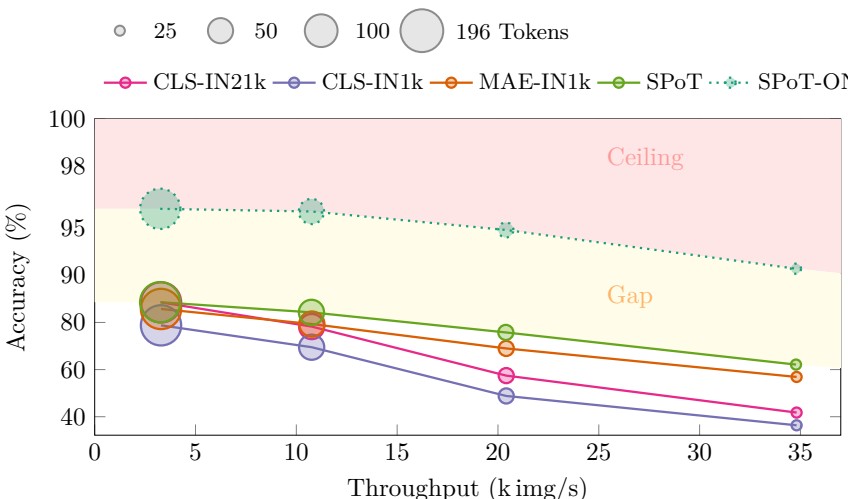

**Figure 5:** We show ImageNet1k accuracy vs. throughput with 5 models at four sparsity levels. The **ceiling** denotes performance unlikely to be achieved given the intrinsic label noise in ImageNet (Beyer et al., 2020). The **gap** highlights the margin between SPoT with optimal configuration and SPoT-ON, illustrating possible performance gain through better token placement.

## 5.1 Robustness and Sensitivity Analysis

To thoroughly validate the semantic relevance of optimized subpixel token placements, we conduct targeted robustness analyses. Specifically, we evaluate performance under intentionally adversarial conditions—including inverse priors favoring irrelevant regions such as backgrounds or image boundaries, and gradient ascent adversarially maximizing loss or randomized label assignments. Our results in Table 6 demonstrate substantial performance degradation in all these adversarial scenarios, strongly suggesting that our token placement mechanism indeed leverages meaningful semantic cues rather than trivial spatial correlations or easily exploitable priors.

Importantly, the gradient–ascent oracle still receives the correct labels, so its sharp accuracy collapse shows that semantically aligned token positions are indispensable—simply keeping the right supervision is not enough if the tokens are pushed onto irrelevant regions. In contrast, the near-chance performance under label-obfuscation demonstrates that the model does not easily adapt to arbitrary image–label pairings, confirming that our selector grounds predictions in genuine object evidence rather than flexible location–label shortcuts. Background sampling leads to reduced performance; however, we posit that the token set still captures some object edges, providing the model with useful information. This hypothesis is reinforced by the significantly larger performance drop observed when sampling with a strong bias toward image boundaries. Adversarial optimization shows a similarly extreme drop, indicating that good token placements are not trivial.

## 5.2 Retrofitting and Finetuning Details

Table 7 details the training configurations for the three backbone variants: CLS-IN1k, CLS-IN21k, and MAE-IN1k. By *retrofitting*, we mean a fine-tuning protocol where the goal is to adapt a models pretrained parameters $\theta$ to our proposed subpixel tokenization scheme, i.e.

$$\min_{\theta} \mathbb{E}_{I \sim \mathcal{I}}[\mathcal{L}(g_\theta(I, S), y)] \text{ s.t. } S = \{s_k \sim \mathcal{U}(\Omega_{\text{subpix}}) : k = 1, ..., m\} \text{ i.i.d. for each image } I.$$

Explicitly, we let $\mathbb{E}_{I \sim \mathcal{I}}$ denote the expectation over a dataset of images $I \sim \mathcal{I}$ of the loss function $\mathcal{L}$ of the model outputs $g_\theta(I, S)$ and the labels $y$. The positions $S$ consists of $m$ i.i.d. uniformly distributed subpixel positions for each image $I$, such that each position $s_k \sim \mathcal{U}(\Omega_{\text{subpix}})$.

The official TIMM (Wightman, 2019) model card names are listed below each corresponding subtable for reference. All protocols use isotropic sampling and run for 50 epochs on $224 \times 224$ images on the ImageNet-

**Table 7:** Training protocols for retrofitting. We use training protocols from He et al. (2022) in MAE finetuning.

**(a)** CLS-IN1k Retrofitting

| config | value |
|---|---|
| sampler | isotropic |
| batch size | 2048 |
| epochs | 50 |
| dataset | ImageNet1k |
| img.size | $224 \times 224$ |
| loss fn. | CE (0.1 smooth.) |
| optimizer | AdamW |
| momentum | 0.9, 0.99 |
| lr.sched. | cos.decay (5 w.u.) |
| lr | 6e−5 |
| dropout path | 0.1 |
| opt. $\epsilon$ | 1e−8 |
| augment | rrc / randaug(15, .5) |
| mixup $\alpha$ | 0.8 |
| cutmix $\alpha$ | 1.0 |
| llrd | 0.65 |

vit_base_patch16_224.augreg_in1k

**(b)** CLS-IN21k Retrofitting

| config | value |
|---|---|
| sampler | isotropic |
| batch size | 2048 |
| epochs | 50 |
| dataset | ImageNet1k |
| img.size | $224 \times 224$ |
| loss fn. | CE (0.1 smooth.) |
| optimizer | AdamW |
| momentum | 0.9, 0.99 |
| lr.sched. | cos.decay (5 w.u.) |
| lr | 6e−5 |
| dropout path | 0.2 |
| opt. $\epsilon$ | 1e−8 |
| augment | rrc / randaug(15, .5) |
| mixup $\alpha$ | 0.8 |
| cutmix $\alpha$ | 1.0 |
| llrd | 0.65 |

vit_base_patch16_224.augreg2_in21k_ft_in1k

**(c)** MAE-IN1k Retrofitting

| config | value |
|---|---|
| sampler | isotropic |
| batch size | 4096 |
| epochs | 50 |
| dataset | ImageNet1k |
| img.size | $224 \times 224$ |
| loss fn. | MSE |
| optimizer | AdamW |
| momentum | 0.9, 0.95 |
| lr.sched. | cos.decay (5 w.u.) |
| lr | 3e−3 |
| dropout path | 0 |
| opt. $\epsilon$ | 1e−8 |
| augment | rrc |
| mixup alpha | 0.8 |
| cutmix | 1.0 |
| llrd | 0.75 |

mae_vit_base_patch16_in1k

1k dataset (Deng et al., 2009). Layer-wise learning rate decay (LLRD) is employed with a slightly more aggressive parameter in the MAE retrofitting, while the MAE finetuning follows the original protocol outlined by He et al. (2022), with minor exceptions[2].

**Evaluation Protocol.** Our evaluation protocol closely follows existing works (Oquab et al., 2024; He et al., 2022; Zhou et al., 2022). We use bicubic interpolation and a crop ratio of 0.875 in our evaluations. All models are trained with standard ImageNet normalization, noting that the TIMM baselines (Wightman, 2019) adopt the convention of using a flat normalization of $\mu_{\text{RGB}} = \sigma_{\text{RGB}} = (0.5, 0.5, 0.5)$. Our kNN evaluation protocol was adapted from Caron et al.'s (2021) work.

# 6 Related Work

Leveraging sparsity to reduce computational overhead is a well-established research direction. Previous work introduced sparsity through masking during pre-training, in self-supervised (He et al., 2022) and language-supervised contexts (Li et al., 2023). Liu et al. (2023) applied sparsity at the fine-tuning stage by initially upsampling images and subsequently randomly dropping patches, thus enhancing efficiency and reducing computational complexity. Distinct from training-centric sparsity approaches, our work induces sparsity during inference by retrofitting ViTs with a subpixel tokenizer, significantly improving throughput. Another line of research explores inference-time sparsity via selective pruning to either discard (Chen et al., 2023; Rao et al., 2021; Yin et al., 2022) or merge (Bolya et al., 2023) tokens (ToMe) based on different heuristics. In contrast, our approach achieves sparsity by sampling rather than selectively pruning tokens during inference.

We include a comparison with ToMe (Bolya et al., 2023) using their ViT-B/16 models in Table 8. ToMe begins with full token budget and progressively reduces the number of tokens by merging them across transformer layers. In contrast, SPoT performs token reduction at the tokenization stage by sampling a smaller set of tokens and maintains a constant token count throughout the network. For this comparison, we report results using the best-performing SPoT configurations and evaluate performance when sampling 100 tokens, and select ToMe configurations that achieve the highest throughput speed-up, as these most closely match the SPoT performance. SPoT achieves higher throughput improvements while incurring a smaller drop in accuracy.

Recently, other works have explored non-grid based tokenization. One interesting line of research looks to leverage subobject tokenization, which extracts fine-grained segmentations as opposed to square

---

[2]Cosine warmup in lr-scheduler, starting from $1 \times 10^{-7}$ with peak learning rate of $1 \times 10^{-3}$, and gradient clipping set to 3.

**Table 8:** Comparing SPoT with 100 token budget against ToMe. We show throughput improvement (Speed-up) and drop in accuracy ($\Delta$ Acc@1) relative to the full token budget baseline. We use ToMe's officially reported results. † marks that token reduction was applied at the finetuning stage. Otherwise, models apply reduction during inference.

| Model | Speed-up | $\Delta$ Acc@1 |
|---|---|---|
| **CLS-IN21k** | | |
| ToMe | 1.95× | -4.20 |
| SPoT | **3.31×** | **-2.95** |
| **MAE-IN1k** | | |
| ToMe | 1.94× | -4.87 |
| ToMe† | 1.95× | -1.71 |
| SPoT | **3.31×** | **-1.13** |

**Table 9:** Comparing ElasticViT to SPoT in sparse regimes.

| Model | Acc@1 (%) / Number of Tokens | | | | | | | | |
|---|---|---|---|---|---|---|---|---|---|
| | **39** | **59** | **78** | **98** | **118** | **137** | **157** | **176** | **196** |
| ElasticViT | 67.17 | 72.65 | 75.47 | 77.47 | 78.18 | 79.73 | 80.81 | 81.31 | 82.04 |
| SPoT-MAE-IN1k | **71.81** | **77.86** | **80.47** | **81.89** | **82.55** | **83.05** | **83.34** | **83.52** | **83.85** |

patches (Aasan et al., 2024; Chen et al., 2025). Other works apply learnable clustering into the transformer architecture via cross-attention operators (Fan et al., 2024; van Steenkiste et al., 2024), while Nguyen et al. (2025) proposed to tokenize each individual pixel. Deformable patches was first proposed in relation to object detection (Xia et al., 2022), but was further extended to general purpose modeling in ElasticViT (Pardyl et al., 2025), which proposed elastic windows as *local augmentations* in standard classification tasks. These are defined as stochastic patch perturbations in scale, position, and erasure via patch dropout. Put simply, ElasticViT relaxes the traditional patch grid of ViTs by randomly shifting, rescaling, and dropping patches during training.

ElasticViT differs from SPoT in key aspects. First, we relax the discrete grid assumption not by perturbing existing patch positions, but rather by directly sampling arbitrary continuous-valued points within the image. Conversely, ElasticViT uses discrete pixel positions, and does not adapt a continuous subpixel approach. Second, we do not explicitly train our model to handle sparse inputs; instead, our method's inherent robustness to sparse token configurations naturally arises from training on continuously sampled points. Nevertheless, comparing our method to ElasticViT is insightful, as their approach is explicitly trained to handle continuous-valued positions and sparse token scenarios. Table 9 compares SPoT with ElasticViT's officially reported results, demonstrating that SPoT consistently outperforms ElasticViT across all evaluated sparse configurations.

## 7 Conclusion

We proposed SPoT for extracting features at continuous subpixel positions, and used oracle-guided gradient search to probe the nature of optimal token placements and ideal spatial sampling priors. Our case studies showed that the flexibility of continuous off-grid placements improves performance out-of-the-box, especially in sparse token budget settings. SPoT-ON provided an estimate of best-case performance from optimal token placement. Although placements are guided via an oracle, these optimal features *exist independently of how they were discovered*, revealing a performance gap that better informed priors could help bridge. While we focused on analyzing the effects of subpixel tokenization under varying sparsity configurations with different spatial priors, the development of learnable spatial priors is a next step towards narrowing the oracle performance gap.

Concretely, since the oracle derives placements from image-dependent statistics, we foresee that a lightweight "policy network" can be trained as a structured regressor over dense local cues to predict good token placements in a single forward pass. A feasible instantiation would involve a lightweight CNN or MLP operating at patch-grid resolution, taking low-level or early-stage features as input to produce a dense importance map to obtain token placements, e.g. via budget-constrained selection. Potential learning signals include

distillation from SPoT-ON, self-supervised alignment, or joint end-to-end training with the backbone. We emphasize that introducing a policy network is orthogonal to the core contribution of SPoT. The current work intentionally focuses on analytic, interpretable placement to isolate the effect of token geometry.

Our study with SPoT on different spatial priors focused on classification on ImageNet, but optimal placements may vary for different datasets and downstream tasks. Beyond image-level classification, we foresee that SPoT can be useful in tasks that require more explicit spatial reasoning such as object localization and detection, where spatial priors could be adapted to emphasize fine-grained or multi-scale positional structure, potentially improving localization accuracy. Similarly, for video understanding tasks like action recognition and temporal localization, SPoT can be extended to incorporate spatiotemporal priors that jointly encode spatial layout and temporal continuity, encouraging consistent feature alignment across frames. Overall, we expect the design and placement of spatial priors to be task-dependent, and exploring SPoT in temporal prediction settings is a promising direction for future work.

By enabling continuous token positioning, SPoT facilitates gradient-based optimization of token placement, which can be advantageous in resource-constrained environments where sparsification is beneficial. Although we limit our scope to employ an oracle to determine optimal token placements, exploring oracle-independent strategies represents a compelling direction for future research. Specifically, integrating efficient saliency-driven objectives or heuristics during inference could potentially enhance throughput efficiency while maintaining competitive performance compared to models utilizing a full token budget. Further improvements may also be seen by allowing the model to adjust the patch window size dynamically during training. Moreover, while the scope of this work is towards modeling in sparse regimes, our results indicate that continuous subpixel token placements provide a novel research direction for ViTs on a more general level.

## Acknowledgments

This work was funded by RCN (the Research Council of Norway) through Visual Intelligence, Centre for Research-based Innovation (309439). We acknowledge Sigma2 (Project NN8104K) for access to the LUMI supercomputer, owned by the EuroHPC Joint Undertaking, hosted by CSC (Finland) and the LUMI consortium through Sigma2, Norway.

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

## A    Retrofitting and Finetuning Terminology

We distinguish between *retrofitting* and *fine-tuning*, which serve different purposes:

**Retrofitting** adapts a pretrained Vision Transformer to a new tokenizer while preserving the original training objective (supervised or self-supervised). All model parameters are learnable, but we apply layer-wise learning-rate decay where early layers incur a *higher learning rate* than later layers. Retrofitting is performed for a limited number of epochs to align the model with the new tokenization scheme while preserving high-level representations.

**Fine-tuning** adapts a pretrained encoder to a downstream task. In our work, this corresponds to ImageNet classification with a linear head attached. Both the encoder and head are trained jointly using the task-specific objective. We fine-tune the pretrained MAE following standard fine-tuning protocols from the original paper (He et al., 2022).

## B    Retrofitting ablations

In Figure B.1 we show training and validation accuracies during the retrofitting stage for the CLS-IN21k model. We observe that the encoder initially aligns quickly with our tokenizer during the first 5 epochs, and that the validation accuracy plateaus during the last 10 epochs. Based on this result, we retrofit the CLS-IN1k and MAE-IN1k models for 50 epochs as well.

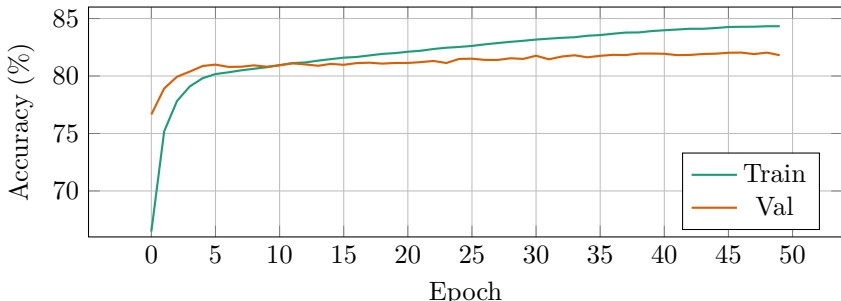

**Figure B.1:** Accuracy per training epoch during retrofitting for our CLS-IN21k model.[3]

## C    Computational overhead of SPoT

SPoT samples $T$ tokens by using bilinear interpolation over a grid defined by the window size $k$. Per point in the grid, bilinear interpolation computes a 4-term weighted sum, requiring 4 multiplications and 3 additions (7 FLOPs; or 4 FLOPs under an FMA-counting convention) for each image channel. Thus, the interpolation cost is $7C$ FLOPs. Computing the interpolation weights incurs an additional constant cost of 8 FLOPs per point. Since this operation is performed for $T \cdot k^2$ points, the total added cost is $T \cdot k^2 \cdot (7C + 8)$ FLOPs. For the standard $k = 16$ window size, $C = 3$ channels, and $T = 25$ tokens, this is 185600 FLOPs added to the tokenization step. We count one multiplication and one addition/subtraction as one FLOP each; non-arithmetic operations (e.g., indexing, clamping, and rounding) are excluded.

More generally, the additional cost of sub-pixel sampling has $O(T \cdot C)$ complexity. In contrast, self-attention and MLP blocks incur $O(T^2 \cdot C)$ and $O(T \cdot C^2)$ complexity, respectively. Therefore, the introduced overhead is asymptotically negligible.

To complement the results in Fig. 5, we show the exact throughput values for SPoT and the patch-based ViTs in Tab. C.1. Despite introducing sub-pixel sampling and interpolation, SPoT closely matches the throughput

---

[3]Validation accuracy is somewhat lower than what is shown in Table 7 due to different validation augmentation schemes. Results in Table 7 follow the standard of resizing to 256 and center cropping to 224, while the accuracies reported during training in Figure B.1 are resized with `RandomResizedCrop(224, scale=(1.0,1.0))` to more closely align with training geometry.

of a standard ViT across all token counts, indicating that the additional operations incur minimal overhead in practice.

**Table C.1:** Throughput (k imgs/s) for standard ViTs and SPoT for different token budgets.

| Model | 25 tok | 50 tok | 100 tok | 196 tok |
|---|---|---|---|---|
| ViT | 34.81 | 20.41 | 10.77 | 3.29 |
| SPoT | 34.78 | 20.39 | 10.75 | 3.25 |

## D  Additional Qualitative Results

We provide additional examples of oracle trajectories over salient image regions for different priors in Fig. D.1. Similar to Fig. 4, they show that the optimized placements do not significantly align with object saliency.

To complement the results in Sec. 5.1, we also provide visualizations of adversarial oracle trajectories with gradient ascent in Figure D.2. This shows that subpixel sampling can also result in reduced performance if adversarial placements are chosen. Similar to the oracle descent case, these placements do not immediately appear good or bad to the human eye, but result in a drastic decrease in performance. In worst-case scenarios, probabilistic priors may accidentally sample token placements that lead to reduced performance.

We note that if the number of oracle iterations is increased to 20 or more, a substantial amount of the token placements move outside the image boundary.

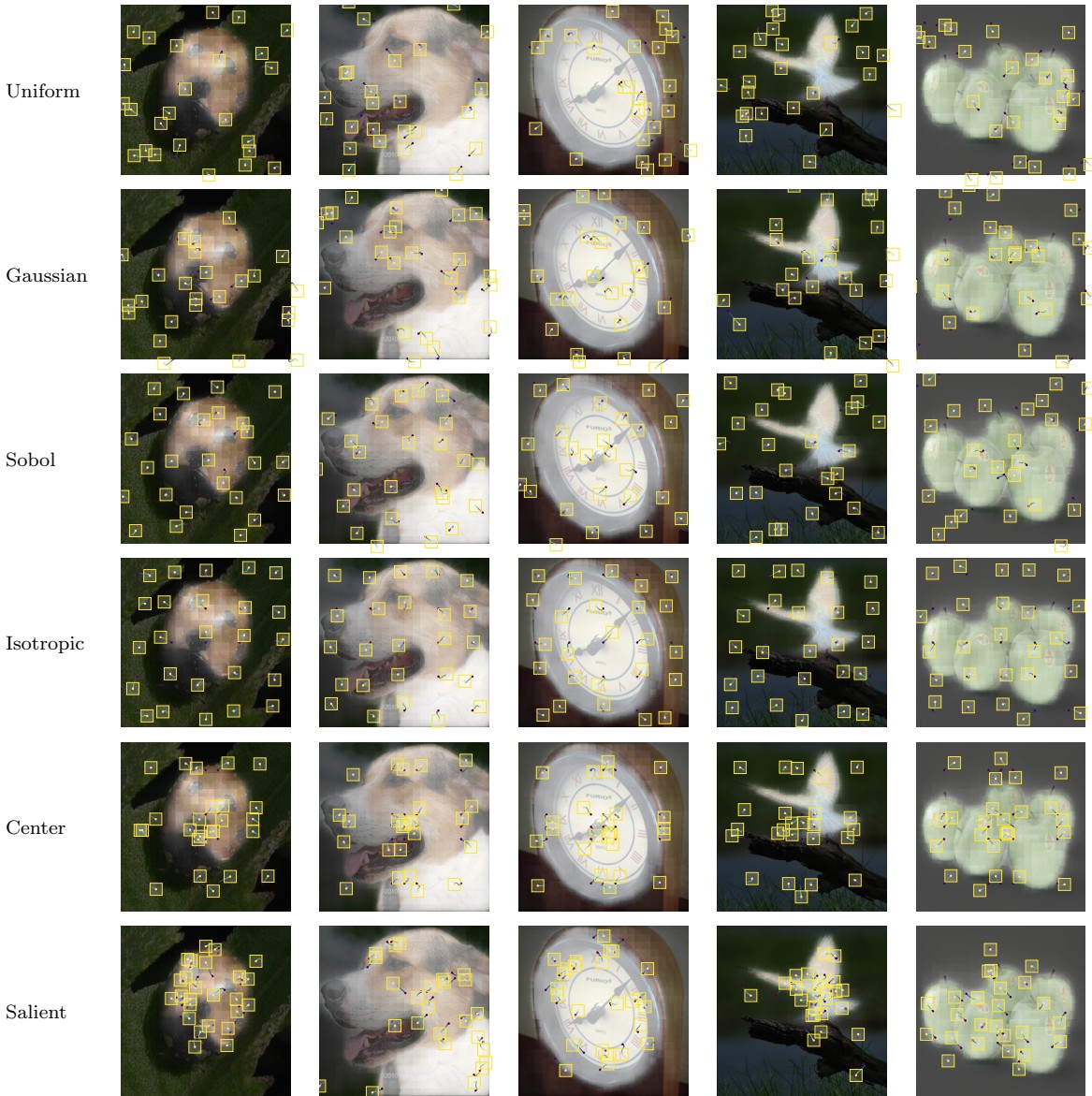

**Figure D.1:** Illustration of oracle placements over salient regions for various priors with 25 tokens using SPoT-ON. Salient image regions are highlighted in white while the background has a dark mask. Trajectories are colored from **dark purple** (start) to **bright yellow** (end).

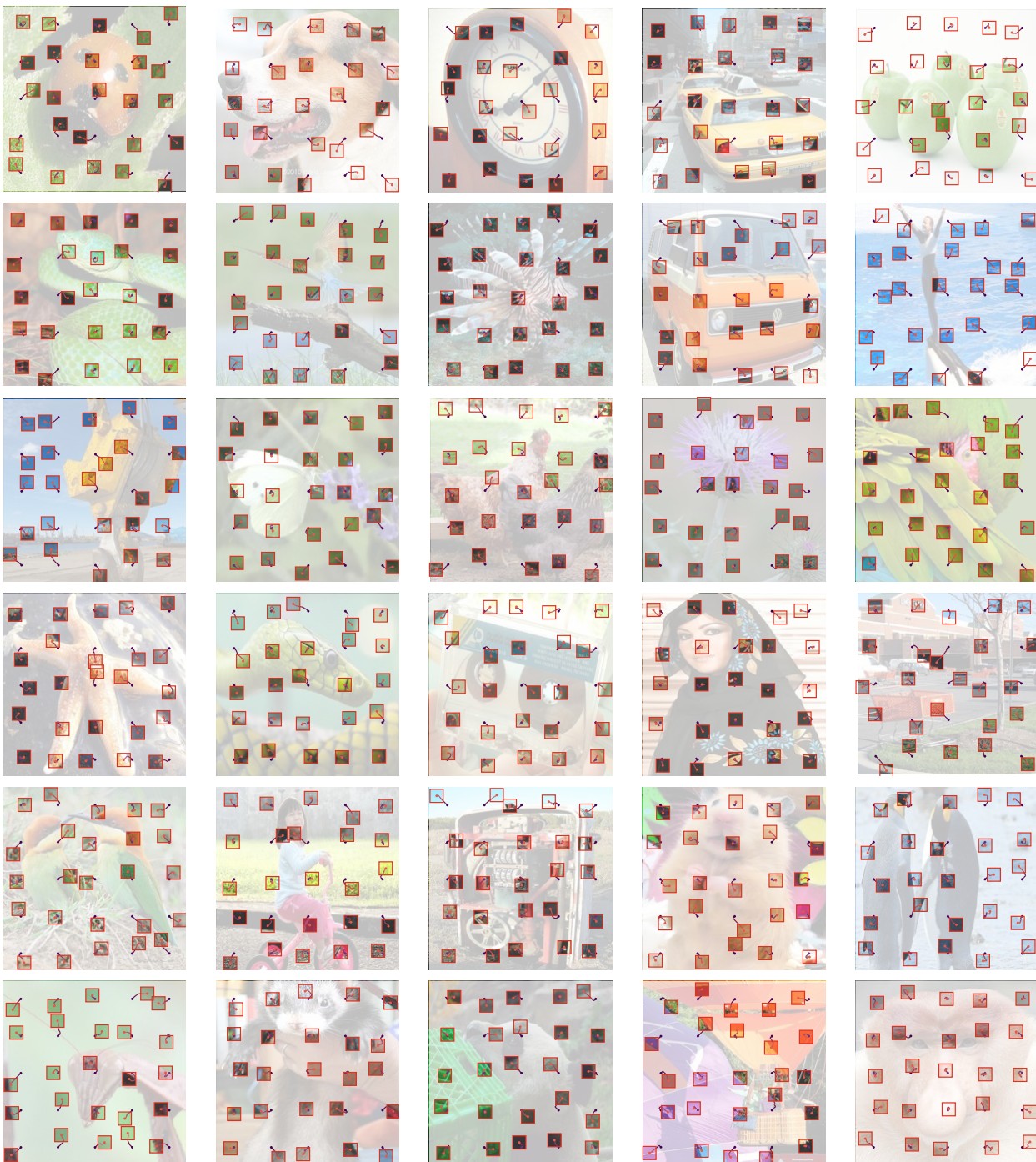

**Figure D.2:** Illustration of *adversarial* oracle placements with 25 tokens using SPoT-ON with gradient *ascent*. Trajectories are colored starting with **dark purple** for initial points, with endpoints colored **red**.

