# OpenReview forum: "SPoT: Subpixel Placement of Tokens in Vision Transformers"
_TMLR — Accepted by TMLR_

### Review · Reviewer_ujJj · 2025-11-11

**Summary Of Contributions:**

The paper presents SPoT, a sub-pixel tokenization framework, where token positions are defined in continuous coordinates rather than discrete grid cells. This formulation enables differentiable optimization of feature locations. The proposed framework has the potential of enhancing the robustness and efficiency of ViTs. To analyze the nature of sparse ViTs and demonstrate the potential of learnable token positions, the paper proposes SPoT-ON, a gradient-based procedure that identifies near-optimal token placements for each image, providing approximate upper bounds on achievable performance in sparse settings. The paper also explores how different priors (including uniform, Gaussian, Sobol, isotropic, center, and saliency-based sampling) affect sparse token performance, revealing that object-centric priors are better in sparse regimes while coverage-based priors are better in dense ones.

**Audience:**

Yes

**Audience Explanation:**

The findings of the paper is highly relevant to the machine learning community, especially researchers working with computer vision and vision transformers. The proposed continuous and differentiable tokenization scheme has the potential of making vision transformers more efficient. The paper also offer insights into sparse vision modeling, helping researchers understand where ViTs optimally look at when constrained by token budgets.

**Broader Impact Concerns:**

N/A.

**Claims And Evidence:**

Yes

**Claims Explanation:**

The paper’s contributions are concisely summarized at the end of the introduction. The authors accurately point out that SPoT-ON is not a practical solution for inference but instead a tool to demonstrate the potential of learnable token positions. The claims made by the paper are supported by empirical studies across several ViT backbones. Results from tables 1-5 show that sub-pixel placement consistently improves performance over all baselines at varying sparsity levels. The paper also includes additional experiment results for robustness and transferability, showing the generalization ability of the learned token placements.

**Requested Changes:**

- Include an analysis on computational overhead: While throughput is briefly mentioned, the cost of sub-pixel sampling and interpolation is not analyzed in detail (e.g., per-image latency, FLOPs). It would be helpful for the readers to judge the potential of the proposed method by including a computational complexity and runtime comparison with baseline ViTs.
- The paper claims that SPoT provides a new direction for interpretable ViT architectures, and the visualizations in Figures 1 and 2 also hints that the sub-pixel patches will be more interpretable and more aligned with the salient objects in the frame. However, the visualizations in Figure 4 does not show any alignment between oracle placements and the salient objects in the frame. As a result I think the paper could be better motivated with some different visualizations as figures 1 and 2. I also encourage the authors to provide more visual examples of saliency maps vs. oracle token positions to better support interpretability claims.
- The authors can discuss the potential applications of SPoT beyond classification (e.g., localization or video understanding).

---

> ### Author Response · Authors · 2025-12-28
> **Response to Reviewer Comment and Requested Changes (1/2)**
>
> Dear reviewer,
>
> We thank you for your helpful comments and suggestions. We provide pointwise responses below. Following TMLR recommendations we will wait with posting an updated manuscript until all 3 reviews have been submitted, but if the reviewer finds it helpful we can submit the changed version earlier.
>
> > Include an analysis on computational overhead: While throughput is briefly mentioned, the cost of sub-pixel sampling and interpolation is not analyzed in detail (e.g., per-image latency, FLOPs). It would be helpful for the readers to judge the potential of the proposed method by including a computational complexity and runtime comparison with baseline ViTs.
>
> We agree that a comparison of computational overhead would be helpful to readers. We briefly show that SPoT has very minimal overhead compared to baselines in Figure 5, however this result would be clearer if a separate table was provided. We will include the following in the revised version of our manuscript.
>
> **Table showing Throughput (k imgs/s):**
> | Model | 25 tok | 50 tok | 100 tok | 196 tok |
> |-------|--------|--------|---------|---------|
> | ViT   |  34.81  | 20.41  | 10.77   | 3.29   |
> | SPoT  |  34.78  | 20.39  | 10.75   | 3.25   |
>
>
> Despite introducing sub-pixel sampling and interpolation, SPoT closely matches the throughput of a standard ViT across all token counts, indicating that the additional operations incur minimal overhead in practice.
>
> More precisely, the additional cost of sub-pixel sampling consists of bilinear interpolation over four neighboring pixels, resulting in an $O(T\cdot C)$ complexity. In contrast, self-attention and MLP blocks incur $O(T^2\cdot C)$ and $O(T\cdot C^2)$ complexity, respectively. Therefore, the introduced overhead is asymptotically negligible. We will also include this in the updated manuscript.
>
> > The paper claims that SPoT provides a new direction for interpretable ViT architectures, and the visualizations in Figures 1 and 2 also hints that the sub-pixel patches will be more interpretable and more aligned with the salient objects in the frame. However, the visualizations in Figure 4 does not show any alignment between oracle placements and the salient objects in the frame. As a result I think the paper could be better motivated with some different visualizations as figures 1 and 2. I also encourage the authors to provide more visual examples of saliency maps vs. oracle token positions to better support interpretability claims.
>
> This is an insightful comment that touches on a very interesting point. Our argument is that SPoT-ON explicitly shows what regions the model "prefers" to produce predictions. Indeed, our expectations were that these would align with saliency maps. As opposed to this, our results show that reality is much more complex, and that the optimized placements actually do not significantly prefer salient regions (Section 4.3 and Figure 4). However, this does not alter the fact that the model has drastically better predictions with SPoT-ON's oracle guided positions than salient ones.
>
> Our results indicate that human-aligned saliency seem different from how ViTs interpret image data. Instead of singling out the object, the model prefers to add context and cues outside the salient object to make their prediction.
>
> We appreciate the reviewer's observation that figure 1 and 2 combined is hinting towards interpretability by object alignment specifically. Figure 2 is meant to illustrate the misalignment problem imposed by grid-based tokenization, but after the reviewer's comment we noticed that the caption says "Grids cannot align every salient region". We think this is the source of confusion, rather than the visualization itself. What we actually want to show here is misalignment over *key features* in the image (whether salient or not). We apologize for this oversight, and will update the caption of Figure 2 in the revised manuscript accordingly.
>
> The patches visualized for SPoT in Figure 1 are based on the observation that object centric *priors* perform better in sparse settings (Section 4.2). In light of this discussion, would it help if we chose a different set of patches for this figure?
>
> Lastly, we interpret the reviewer's request for "more visual examples of saliency maps vs. oracle token positions" as figures showing how oracle trajectories move over the salient regions for salient/non-salient priors. We have uploaded an image to the following anonymous link, for your review: https://anonymous.4open.science/r/spot-anon-DF6C/assets/saliency_vs_oracle.png. We will include these figures in the supplemental of the updated manuscript. Similar to Figure 4, they show that the optimized placements do not significantly align with object saliency--something we can only interpret from the models using SPoT. Please let us know if additional visualizations would be helpful.

---

> ### Author Response · Authors · 2025-12-28
> **Response to Reviewer Comment and Requested Changes (2/2)**
>
> > The authors can discuss the potential applications of SPoT beyond classification (e.g., localization or video understanding).
>
> Thank you for highlighting this. We agree that a discussion of additional potential applications will be helpful for readers. We will include the following discussion in the revised manuscript.
>
> *Beyond image-level classification, we foresee that SPoT may be useful for tasks requiring more explicit spatial reasoning, such as object localization and detection, where spatial priors could be adapted to emphasize fine-grained or multi-scale positional structure and potentially improve localization accuracy. Similarly, for video understanding tasks such as action recognition and temporal localization, SPoT can be extended to incorporate spatiotemporal priors that jointly encode spatial layout and temporal continuity, encouraging consistent feature alignment across frames. Overall, we expect the design and placement of spatial priors to be task-dependent, and exploring SPoT in temporal and other prediction settings is a promising direction for future work.*
>
> Best regards,
> The authors

---

> ### Author Response · Authors · 2026-01-21
> **Revision of Submission**
>
> Dear Reviewer,
>
> Now that all reviews have been posted, we have submitted a revised version of the manuscript. For ease of review, changes addressing your comments are highlighted in **green**.
>
> Thank you for your time and consideration.
>
> Best regards,
> The authors

---

### Review · Reviewer_31UR · 2025-11-15

**Summary Of Contributions:**

This submission proposes a method called SPoT to identify the optimal subpixel patch combination for the label "y" in the pre-trained DiT model. It could serve as a tool to analyze or investigate which key patches we can select in an overlapped way during model inference to achieve better accuracy.

However, the leakage of ground-truth labels leads to the abnormal high accuracy (over 30% improvement on imagenet-1k eval.) and unfair comparison.
Thus, these experimental results can not be used to demonstrate the claim that the proposed SPoT method could surpass and replace the classical image tokenization way.

**Audience:**

Yes

**Audience Explanation:**

This work could serve as an analysis tool for the ViT model, like an extension of the gradient-based saliency map at the token level of the ViT-based model, providing a way to identify the flexible patch combinations that most contribute to classification output.

**Claims And Evidence:**

No

**Claims Explanation:**

Authors challenge the widely-used fixed-grid patchify method with three limitations and claim that their proposed flexible SPoT method addresses these limitations and achieves better accuracy.

However, the methodology design and experiment cannot support their claim due to the following cues:

1. Data leakage and unfair comparison. The label information is used to optimize a better combination, causing the ground truth label 'y' to be leaked, as shown in equation (3), which could definitely get a higher accuracy. This is expected, as the optimization direction during patch selection aligns with the classification goal; lower logit loss is associated with higher accuracy.
2. SPoT is applied on the pre-trained ViT model and cannot be used in model training from scratch in such way.

**Requested Changes:**

I strongly encourage the authors to majorly modify and reorganize the paper to give a clear position for the proposed method. The current submission quite confuses readers about what the paper is exactly doing.

The current version of the submission shows that the authors post a large motivation, but only propose a simple tool method to analyze the pre-trained DiT model with some unsuitable experiment comparisons. For example,
1) authors challenge the widely-used patchify method with three limitations, but their proposed SPoT-ON can not be used for model training and only achieves higher inference accuracy via the ground truth label leakage way.
2) authors post many offline sampling priors in Sec. 3.1 &Figure 3 and recognize the necessity of a learnable prior, but leave this step as future work. etc.

---

> ### Author Response · Authors · 2025-12-28
> **Clarification in Response to Reviewer Comment**
>
> Dear reviewer,
>
> We thank the reviewer for their comments, and hope that the following serves to elucidate our paper. Following TMLR recommendations we will wait with posting an updated manuscript until all 3 reviews have been submitted, but if the reviewer finds it helpful we can submit the changed version earlier.
>
> We would like to start by addressing some potentially significant misunderstandings.
>
> **We wish to clarify  that our results are for traditional ViT models; at no point is our work discussing "DiTs".** If the reviewer is referring to Diffusion Transformers, we note that the diffusion literature uses the term "tokenization" to refer to the encoding of latent representations in an encoder/decoder framework. This is different from standard tokenization which refers to the conversion of high dimensional inputs (e.g. images) into a sequence of embeddings used as input to a transformer. Our work is about the latter.
>
> We also want to emphasize the difference between our tokenization method SPoT, and our analysis tool Oracle Neighborhood search (SPoT-ON). In the revised manuscript, we will make it more explicit in our list of contributions (Section 1) that SPoT-ON is a tool for analysis, and can not be used in practice during inference.
>
> **SPoT is trainable**, does *not rely on an oracle* (**no data leakage**), and **outperforms baselines** across all sparsity levels. SPoT addresses the problems of interdependence, combinatorial search, and misalignment in *sparse feature sampling* for standard grid-based ViTs by redefining the feature extraction method.
>
> Meanwhile, **we use SPoT-ON as an analysis tool** to get insights about the nature of optimal token placements. Our case studies in Section 4 provide four key insights ("findings") about token placements in sparse settings using SPoT-ON to probe when the models perform best. A key thing to note is that the placements found by the oracle could simply be sampled by chance. What the oracle shows is that *optimal placements exist* that allow ~30\% improvement in accuracy while the *parameters of the model are unchanged*. The leakage of the labels in the oracle is used to theorize on what we can achieve by only *changing what the model sees*, and what the characteristics of optimal token placements are. In other words, SPoT-ON is used as a tool to get approximate upper bounds on achievable performance with SPoT.
>
> We will incorporate this explanation of the difference between SPoT and SPoT-ON in the revised manuscript, and thank the reviewer for improving the clarity of the paper.
>
> We will address each of the reviewer's concerns in a follow up comment.
>
> Best regards,
> The authors

---

> ### Author Response · Authors · 2025-12-28
> **Response to Reviewer Concerns**
>
> Below we address each of the reviewer's concerns.
>
> > Data leakage and unfair comparison. The label information is used to optimize a better combination, causing the ground truth label 'y' to be leaked, as shown in equation (3), which could definitely get a higher accuracy. This is expected, as the optimization direction during patch selection aligns with the classification goal; lower logit loss is associated with higher accuracy.
>
> *We want to make it absolutely clear that our Oracle-guided Neighborhood search (SPoT-ON) is a tool for analyzing model behavior*, and the corresponding results only show the *potential* performance gain that is possible through better token placements. As the reviewer has noted, the SPoT-ON results can not and should not be directly compared to baselines to make claims about currently achievable performance improvements. We have highlighted this several places in the text, for example:
> - "We specify that SPoT-ON incurs a higher computational cost for classification, and is not intended as a practical solution for inference. Rather, *it acts as a tool for analyzing the nature of sparse ViTs, demonstrating the potential of learnable token positions*." (Section 3.2, where the oracle is introduced)
> - "The gap highlights the margin between SPoT with optimal configuration and SPoT-ON, illustrating possible performance gain through better token placement." (Figure 5 caption)
> - "SPoT-ON provided an estimate of best-case performance from optimal
> token placement. Although placements are guided via an oracle, these optimal features exist independently of how they were discovered, revealing a performance gap that better informed priors could help bridge." (Section 7)
> - "\[...\] the development of learnable spatial priors is a next step towards narrowing the oracle performance gap." (Section 7)
> - We consistently label these results as "oracle" or "SPoT–ON" results in all tables and captions, to make clear that oracle guidance was used. In figure 5 we show the SPoT-ON results with a faint dotted line to visually separate it from the baselines and our SPoT results, which do not rely on oracle guidance.
>
> In addition, *we will make it more explicit in our list of contributions in Section 1 that SPoT-ON is a tool for analysis and can not be used for inference in practice*. Are there other places we could make this clearer in the text?
>
> > \[...\] these experimental results can not be used to demonstrate the claim that the proposed SPoT method could surpass and replace the classical image tokenization way.
>
> Unlike SPoT-ON, our standard method SPoT does ***not*** rely on the oracle ("data leakage"). Instead, the performance improvements of SPoT over standard ViTs are a result of greater flexibility by allowing subpixel placements and appropriate selection of sampling priors. We refer the reviewer to the results of Table 5 and Figure 5, which show that *SPoT outperforms baselines with classical tokenization across all sparsity levels*.
>
>
> > SPoT is applied on the pre-trained ViT model and cannot be used in model training from scratch in such way.
>
> We *retrofit* three pre-trained ViT models for 50 epochs to integrate them with SPoT; the training details are provided in Section 5.2. Put simply, we replace the classical ViT tokenizer with SPoT, and finetune with same loss that was used for pretraining to remove any bias towards grid-alignment of patches incurred by the pretraining. It is entirely possible to train a ViT with SPoT from scratch. If the reviewer is referring to Oracle Neighborhood search (SPoT-ON), it is an analysis tool for evaluation of token placements, where everything except patch positions are frozen, and no actual model training is involved.
>
> > This work could serve as an analysis tool for the ViT model, like an extension of the gradient-based saliency map at the token level of the ViT-based model, providing a way to identify the flexible patch combinations that most contribute to classification output.
>
> We agree that this is an important point; and this was indeed our intention with our SPoT-ON oracle guided experiments. We emphasize this analysis (Section 4) as one of our key contributions and insights.
>
> > The current submission quite confuses readers about what the paper is exactly doing.
>
> It would be very helpful if the reviewer could point out the specific sections that could confuse readers. We wish to make our work and contributions as clear as possible.
>
> Best regards,
> The authors

---

> ### Author Response · Authors · 2025-12-28
> **Response to Changes Requested by the Reviewer**
>
> Below we address the requested changes.
>
> > [RC1] "authors challenge the widely-used patchify method with three limitations, but their proposed SPoT-ON can not be used for model training and only achieves higher inference accuracy via the ground truth label leakage way."
>
> We wish to clarify that our proposed method SPoT *can* be used for model training, while the Oracle Neighborhood search (SPoT-ON) is only an evaluation and analysis tool. Further, ***SPoT achieves higher inference accuracy than the baselines across all sparsity levels -- see Table 5 and Figure 5***. This performance increase is achieved by optimal selection of (offline) sampling priors.
>
> > [RC2] "authors post many offline sampling priors in Sec. 3.1 &Figure 3 and recognize the necessity of a learnable prior, but leave this step as future work. etc."
>
> The sampling priors provide different ways for sparse token selection from the get-go, something that standard grid-based tokenization does not allow. We think that a learnable prior may help close the performance gap of the potentially achievable performance demonstrated by the oracle evaluation tool SPoT-ON, but it is *not necessary* for improving performance as *SPoT already outperforms baselines*.
>
> This paper focuses on the development of the SPoT tokenization method and offering several insights into optimal tokenization strategies in sparse settings. We agree that learnable priors for SPoT are extremely relevant, but consider it outside the scope of the current paper. Initial ideas for moving in this direction have not yet yielded results that outperform the current SPoT.
>
>
> Best regards,
> The authors

---

> ### Author Response · Authors · 2026-01-21
> **Revision of submission**
>
> Dear Reviewer,
>
> Now that all reviews have been posted, we have submitted a revised version of the manuscript. For ease of review, changes addressing your comments are highlighted in **purple**.
>
> We would appreciate your feedback on whether these revisions adequately address the concerns raised in your review.
>
> Best regards,
> The authors

---

> > ### Comment · Reviewer_31UR · 2026-01-21
> > **Any following reply plan?**
> >
> > Thanks for your reply.
> >
> > I have reviewed the revised submission. There are only two sentences highlighted in purple; no changes to the main content in the methods and experiments.
> >
> > The current revision is not adequate to address my concerns. I maintain my original request of a Major Revision
> >
> > I am wondering whether authors plan to provide point-by-point responses to the questions I listed, as in the reply to Reviewer uzzG.

---

> > > ### Author Response · Authors · 2026-01-21
> > >
> > > We apologize for the confusion.  We replied to your review, but noted that our response was not visible to everyone. Please see our response for details.

---

> > > > ### Comment · Reviewer_31UR · 2026-01-22
> > > > **Reply to authors' response**
> > > >
> > > > I appreciate the authors' detailed responses. Sorry for my typo and confusion, all 'DiT' in my first comment refers to 'ViT'.
> > > >
> > > > The responses address some of my questions and help me better understand the paper. I have some suggestions for further revision.
> > > >
> > > > 1. **Please do not exaggerate the effect of SPoT-ON (based on label leak).** It is not surprising that tokens/regions selected under the supervision of the ground-truth label can achieve super high top-1 accuracy. Tables 1 and 2 confuse the reader into thinking that your proposed method can achieve over 30% improvement on ImageNet. It is great to show the gap provided by the ON in Figure 5. This gap is identified as a potential improvement in the future but not filled in this paper.
> > > >
> > > > 2. **The proposed SPoT is trivial.** The model-free spatial priors tested in the paper are not hard to come up with, and the saliency method has been proposed before.
> > > >
> > > > 3. **There is no baseline method in the paper.** In my understanding, there are only two kinds of fair comparison results: base-ViT and finetuned SPoT-ViT. It is also not surprising that finetuned ViT outperforms the original one. There are many Adaptive Token or Token Pruning papers that can be chosen as baselines.
> > > >
> > > > 4. **Retrofitting vs. Finetuning**. Which parameters are learnable in the retrofitting and fine-tuning? What's the difference between these two training configurations?

---

> > > > > ### Author Response · Authors · 2026-01-25
> > > > > **Reply to reviewer's comment (1/2)**
> > > > >
> > > > > Dear reviewer,
> > > > >
> > > > > Thank you for reviewing our response. We address each of your points below.
> > > > >
> > > > > > Please do not exaggerate the effect of SPoT-ON. It is not surprising that tokens/regions selected under the supervision of the ground-truth label can achieve super high top-1 accuracy. Tables 1 and 2 confuse the reader into thinking that your proposed method can achieve over 30% improvement on ImageNet. It is great to show the gap provided by the ON in Figure 5. This gap is identified as a potential improvement in the future but not filled in this paper.
> > > > >
> > > > > We understand your concern regarding the potential of misinterpreting results, and fully agree on the importance of clear communication of the nature of our results.
> > > > >
> > > > > *SPoT-ON is intentionally described as an oracle*, following established oracle-based and idealized-benchmark analyses in the learning and optimization literature, where access to optimal (but generally unattainable) solutions is assumed to isolate and study the effect of a specific factor (in our case, token sampling geometry) independently of the mechanism used to obtain it. This terminology is intended to emphasize that these results represent an upper bound rather than achievable performance.
> > > > >
> > > > > We agree that the presentation can be clarified further. Following your suggestion we will update the captions of Table 1 and 2 to highlight the use of the oracle even further. We will also add a dedicated column in Table 1 labeling SPoT-ON as results as "Oracle" to distinguish them from the non-oracle SPoT result (Table 2 already has an Oracle column).
> > > > >
> > > > > *Table 1: Oracle accuracy of grid-constrained and off-grid patch representations in extreme sparse setting with 12.5% of tokens. The grid-based configuration mimics the discrete patch selection of standard ViTs. The off-grid configuration permits subpixel placement in continuous space. Results demonstrate that allowing continuous positioning enhances representational quality under sparse token regimes. We report the performance of initially sampled points (SPoT) and oracle optimized placements (SPoT-ON).*
> > > > >
> > > > > *Table 2: Accuracy (\%) for different spatial initialization priors in extreme sparse setting with 25 tokens. We show SPoT performance (Acc@1) and the potential increase in performance obtained under oracle optimization using SPoT-ON (Oracle $\Delta$).*
> > > > >
> > > > > Please let us know if there are additional places where the distinction between oracle and non-oracle results could be made clearer.
> > > > >
> > > > >
> > > > > > The proposed SPoT is trivial. The model-free spatial priors tested in the paper are not hard to come up with, and the saliency method has been proposed before.
> > > > >
> > > > > We agree that individual spatial priors such as uniform, Gaussian, or center-biased distributions are conceptually simple when considered in isolation. However, the priors themselves are not intended to be the core contribution of this work.
> > > > >
> > > > > Our proposed method is the **SPoT subpixel tokenization framework** (and SPoT-ON for analysis), which enables a systematic and controlled analysis of token sampling strategies that is not possible with standard patch tokenizers. In particular, SPoT reformulates tokenization as a **continuous sampling procedure**, allowing token locations to lie between pixel centers. This is essential both for sampling from continuous probability maps (e.g., uniform, Gaussian, saliency-based) and for deterministic priors (e.g., isotropic or center-biased) that place tokens at non-integer locations.
> > > > >
> > > > > This continuous formulation is what enables the oracle analysis: optimal token placements can be optimized via gradient-based methods only because token locations are differentiable. With conventional, grid-aligned patch tokenization, such an investigation of sampling priors, and their interaction with model performance, would not be feasible.
> > > > >
> > > > > Importantly, while the individual priors may be trivial, their **systematic evaluation within a unified, subpixel tokenization framework** constitutes a contribution in itself. The analysis reveals how different priors affect downstream performance under controlled conditions, thereby clarifying the role of sampling geometry.
> > > > >
> > > > > From a technical standpoint, SPoT is not a trivial modification: it requires reframing tokenization as movable patch sampling, interpolating underlying pixel values for feature extraction, and introducing kernelized positional embeddings to generalize standard patch-based positional encodings.
> > > > >
> > > > > If there exist prior works that propose comparable subpixel tokenization mechanisms or conduct a similarly controlled analysis of token sampling priors (including saliency-based ones) within Vision Transformers, we would welcome pointers and would be happy to discuss the relationship.

---

> > > > > > ### Author Response · Authors · 2026-01-25
> > > > > > **Reply to reviewer's comment (2/2)**
> > > > > >
> > > > > > > There is no baseline method in the paper. In my understanding, there are only two kinds of fair comparison results: base-ViT and finetuned SPoT-ViT. It is also not surprising that finetuned ViT outperforms the original one. There are many Adaptive Token or Token Pruning papers that can be chosen as baselines.
> > > > > >
> > > > > > Our retrofitting strategy is intentionally designed to avoid unfair finetuning; we use an aggressive layer-wise learning rate decay (llrd) to update the initial ViT layers to align with the new tokenizer while keeping the deeper layers similar, and only retrofit for 50 epochs (details in Table 7).
> > > > > >
> > > > > > We compare with the standard grid-based ViTs as baselines for each of the models (CLS-IN1k, CLS-IN21k, MAE-IN1k). These are the models we retrofitted SPoT from, and therefore serve as the natural baselines.
> > > > > >
> > > > > > From section 5:
> > > > > > *We present the performance of SPoT under varying sparsity configurations and compare it against baseline models, including the supervised backbones from TIMM (Wightman, 2019 and the officially fine-tuned MAE (He et al., 2022).
> > > > > > For clarity, all baselines are denoted as ViT-B/16 in Table 5. To evaluate the baselines under sparsity constraints, we apply PatchDropout (Liu et al., 2023), which randomly drops input patches during inference.*
> > > > > >
> > > > > > Our paper studies whether grid-based tokenization is an inherent limitation in ViTs, and whether subpixel placement of tokens has the potential to improve the standardized vision transformer architecture by providing more flexibility and control. Unlike prior adaptive tokenization methods that prune, merge, or reweight tokens **after** fixed-grid tokenization, SPoT adapts token placement itself in continuous space, enabling principled analysis of sampling geometry.
> > > > > >
> > > > > > Nevertheless, we provide a comparison against the token merging method ToMe [1]. We will include the following result in the revised manuscript.
> > > > > >
> > > > > > *We include a comparison with ToMe (Bolya et al., 2023) using their ViT-B/16 models in Table 8. ToMe begins with full token budget and progressively reduces the number of tokens by merging them across transformer layers. In contrast, SPoT performs token reduction at the tokenization stage by sampling a smaller set of tokens and maintains a constant token count throughout the network. For this comparison, we report results using the best-performing SPoT configurations and evaluate performance when sampling 100 tokens, and select ToMe configurations that achieve the highest throughput speed-up, as these most closely match SPoT performance. SPoT achieves higher throughput improvements while incurring a smaller drop in accuracy.*
> > > > > >
> > > > > > *Table 8: Comparing SPoT with 100 token budget against ToMe. We show throughput improvement (Speed-up) and drop in accuracy (∆ Acc@1) relative to the full token budget baseline. We use ToMe's officially reported results. † marks that token reduction was applied at the finetuning stage. Otherwise, models apply reduction during inference.*
> > > > > >
> > > > > > | Model | Method | Speed | $\Delta$ Acc\@1|
> > > > > > |-------|--------|---------|---------|
> > > > > > | CLS-IN21k | ToMe   |  1.95$\times$     | -4.20% |
> > > > > > | CLS-IN21k | SPoT  | 3.31$\times$  | -2.95% |
> > > > > > | MAE-IN1k  | ToMe |  1.94$\times$   | -4.87%|
> > > > > > | MAE-IN1k  | ToMe  † |  1.95$\times$   | -1.71%
> > > > > > |MAE-IN1k   | SPoT |    3.31$\times$  |  -1.13%   |
> > > > > >
> > > > > > The SPoT results are based on the best configuration (without oracle) from table 5.
> > > > > >
> > > > > >
> > > > > > > Retrofitting vs. Finetuning. Which parameters are learnable in the retrofitting and fine-tuning? What's the difference between these two training configurations?
> > > > > >
> > > > > > **Retrofitting vs. Fine-tuning.**
> > > > > > We use the terms retrofitting and fine-tuning to denote two distinct training configurations that differ in their objective and purpose.
> > > > > >
> > > > > > **Retrofitting.**
> > > > > > Retrofitting refers to adapting a pre-trained Vision Transformer to a new tokenizer while preserving the original training objective, whether supervised or self-supervised. All network parameters remain learnable. To facilitate alignment with the new tokenization scheme while limiting changes to high-level representations, we apply aggressive layer-wise learning rate decay, such that earlier layers are updated more strongly than later layers. Model-specific training configurations are provided in Table 7.
> > > > > >
> > > > > > **Fine-tuning.**
> > > > > > Fine-tuning refers to adapting a pre-trained encoder to a new downstream task. In our case, this corresponds to fine-tuning a SPoT-MAE for ImageNet classification. A linear classification head is attached, and both the head and the encoder are trained jointly using the classification objective. All network parameters are learnable, and we follow the fine-tuning protocol and hyperparameters used in the original MAE paper.
> > > > > >
> > > > > > We hope this clarification clearly distinguishes the two training configurations.
> > > > > >
> > > > > > Best regards, The authors
> > > > > >
> > > > > >
> > > > > > [1] D. Bolya, C.-Y. Fu, X. Dai, P. Zhang, C. Feichtenhofer, and J. Hoffman, “Token merging: Your ViT but faster,” in Inter. Conf. Learn. Represent. (ICLR), 2023.

---

> > > > > > > ### Comment · Reviewer_31UR · 2026-01-25
> > > > > > > **Reply**
> > > > > > >
> > > > > > > Thanks for the detailed reply. My concerns have been addressed.
> > > > > > >
> > > > > > > Some listed clarifications should be appended to the submission in order to make it clearer for the reader.
> > > > > > >
> > > > > > > For example, when I first saw 'Retrofitting', which is not a common term in Computer Vision, I thought it was fine-tuning the tokenizer (PatchEmbed module) or the first several layers. But all layers are actually set to learnable, as clarified in the reply.

---

> > > > > > > > ### Author Response · Authors · 2026-01-27
> > > > > > > > **Final revision**
> > > > > > > >
> > > > > > > > Dear reviewer,
> > > > > > > >
> > > > > > > > Thank you for your constructive feedback and for helping us improve the clarity of the manuscript.
> > > > > > > > Following your suggestion, we have made a final revision and added a short appendix section clarifying the terminology and training configurations discussed in our response.
> > > > > > > >
> > > > > > > > Best regards,
> > > > > > > > The authors

---

### Review · Reviewer_uzzG · 2026-01-12

**Summary Of Contributions:**

This paper challenges the conventional reliance on discrete, grid-aligned tokenization in Vision Transformers (ViTs). The authors argue that forcing tokens into a fixed grid is sub-optimal for sparse feature selection, as critical image patterns often straddle grid lines. To resolve this, they introduce Subpixel Placement of Tokens (SPOT), a framework that allows tokens to occupy continuous coordinates within the image space. By leveraging a bilinear interpolation function, the token positions become differentiable, enabling gradient-based optimization. The authors propose SPOT-ON, an oracle-guided search that identifies optimal token locations per image. Furthermore, the paper investigates various spatial priors, demonstrating that object-centric biases significantly enhance performance in sparse regimes, and that optimized placements are transferable across independently trained models.

Key Strengths

1. The work elegantly re-frames the ViT as a "visual bag-of-words" by decoupling tokenization from the rigid pixel grid, providing a more flexible inductive bias for sparse modeling.

2. The empirical evidence shows that SPOT-ON can achieve 90.9% accuracy on ImageNet-1k with only 12.5% of tokens (25 tokens), revealing a massive performance "ceiling" that current sparse methods fail to reach.

3. In high-sparsity settings, the method outperforms grid-constrained approaches by over 16.9, highlighting the inherent advantage of subpixel alignment.

4. The finding that token placements optimized by one model can improve the performance of another model suggests that SPOT-ON captures fundamental semantic structures rather than model-specific noise.

Key Weaknesses

1. The primary performance gains stem from SPOT-ON, which requires iterative gradient-based optimization during inference. This makes the most powerful version of the proposed method computationally prohibitive for real-time applications.

2. There is a substantial disparity between "out-of-the-box" spatial priors and the oracle-guided results. The paper lacks a practical, lightweight mechanism (e.g., a predictor network) to bridge this gap without iterative optimization.

3. The method requires a 50-epoch "retrofitting" phase on ImageNet-1k to adapt pre-trained weights to the subpixel tokenizer. While shorter than pre-training, it remains a non-trivial cost for large-scale models.

4. The scope is limited to image classification. It remains unproven whether subpixel flexibility offers similar advantages for dense prediction tasks like object detection or segmentation, where spatial precision is traditionally handled by grid-based heuristics.

**Audience:**

Yes

**Audience Explanation:**

This work addresses fundamental questions regarding ViT architecture and efficiency. Researchers focused on efficient AI, Transformer inductive biases, and spatial representation will find the "visual bag-of-words" perspective highly valuable.

**Broader Impact Concerns:**

The paper focuses on technical efficiency and architectural improvements for AI reliability. There are no significant ethical or social impact concerns beyond the standard biases inherited from large-scale datasets like ImageNet.

**Claims And Evidence:**

Yes

**Claims Explanation:**

The authors provide multi-faceted evidence, including direct comparisons between grid and subpixel optimization in identical settings. The robustness tests using adversarial oracles and transferability experiments further confirm that the model leverages genuine semantic cues.

**Requested Changes:**

1. Provide a detailed quantitative analysis comparing the FLOPs/latency added by the bilinear interpolation step in SPOT vs. standard linear patch projection.

2. The authors mention learnable priors as future work. Please expand this into a concrete discussion on the feasibility of training a lightweight "policy network" to predict SPOT-ON placements in a single forward pass.

3. Include an ablation study showing how performance scales with the number of retrofitting epochs (e.g., results after 5, 10, and 25 epochs) to identify the minimum adaptation required.

4. Supplement the qualitative results with examples where SPOT-ON fails to improve performance, helping the audience understand the limits of subpixel sampling.

---

> ### Author Response · Authors · 2026-01-21
> **Reply to Reviewer Comment and Requested Changes**
>
> We thank the reviewer for the thorough consideration of our paper. Below we provide point-wise responses to the requested changes. For clarity, we have marked the reviewer's requested changes in **blue** in the updated manuscript.
>
> > Provide a detailed quantitative analysis comparing the FLOPs/latency added by the bilinear interpolation step in SPOT vs. standard linear patch projection.
>
> We agree that this is important information, and that a more detailed analysis would help demonstrate that the proposed tokenizer introduces very minimal computational overhead.
>
> In addition to the throughput table and complexity analysis added in response to reviewer ujJj, we have included the following paragraph in appendix B on computational overhead in the revised manuscript.
>
> *SPoT samples $T$ tokens by using bilinear interpolation over a grid defined by the window size $k$. Per point in the grid, bilinear interpolation computes a 4-term weighted sum, requiring 4 multiplications and 3 additions (7 FLOPs; or 4 FLOPs under an FMA-counting convention) for each image channel. Thus, the interpolation cost is $7C$ FLOPs. Computing the interpolation weights incurs an additional constant cost of $8$ FLOPs per point. Since this operation is performed for $T\cdot k^2$ points, the total added cost is $T\cdot k^2 \cdot (7C + 8)$ FLOPs. For the standard $k=16$ window size, $C=3$ channels, and $T=25$ tokens, this is $185 600$ FLOPs added to the tokenization step. We count one multiplication and one addition/subtraction as one FLOP each; non-arithmetic operations (e.g., indexing, clamping, and rounding) are excluded.*
>
> > The authors mention learnable priors as future work. Please expand this into a concrete discussion on the feasibility of training a lightweight "policy network" to predict SPOT-ON placements in a single forward pass.
>
> We have included the following paragraph in the conclusion of the revised article, and hope that it clarifies the direction of future work.
>
> *Concretely, since the oracle derives placements from image-dependent statistics, we foresee that a lightweight "policy network" can be trained as a structured regressor over dense local cues to predict good token placements in a single forward pass. A feasible instantiation would involve a lightweight CNN or MLP operating at patch-grid resolution, taking low-level or early-stage features as input to produce a dense importance map to obtain token placements, e.g. via budget-constrained selection. Potential learning signals include distillation from SPoT-ON, self-supervised alignment, or joint end-to-end training with the backbone.
> We emphasize that introducing a policy network is orthogonal to the core contribution of SPoT. The current work intentionally focuses on analytic, interpretable placement to isolate the effect of token geometry.*
>
>
> > Include an ablation study showing how performance scales with the number of retrofitting epochs (e.g., results after 5, 10, and 25 epochs) to identify the minimum adaptation required.
>
> We agree that this is an important ablation, and have included a figure showing the train and validation accuracies during the retrofitting stage in appendix A of the updated manuscript.
>
>
> > Supplement the qualitative results with examples where SPOT-ON fails to improve performance, helping the audience understand the limits of subpixel sampling.
>
>
> We interpret this request as a figure showing the adversarial SPoT-ON placements using gradient ascent, which shows that subpixel sampling can also result in reduced performance if adversarial placements are chosen.
>
> This qualitative visualization nicely complements the adversarial ablations in section 5.1. We have included the figure in the appendix, section C. Similar to the oracle descent case, these placements do not immediately appear good or bad to the human eye, but result in a drastic decrease in performance as shown in table 6. This means that in worst-case scenarios, probabilistic priors may accidentally sample token placements that lead to reduced performance.
>
> We hope that this satisfies the reviewer's requests, and addresses the weaknesses mentioned in the review. We have made a revision to the submission with the updated manuscript.
>
> Best regards,
> The authors

---

### Decision · Action_Editor_ar9h · 2026-02-11

**Recommendation:** Accept with minor revision

**Additional Comments:**

The authors should clarify how the model parameters $\theta$ are optimized when using SPoT. In particular, it is unclear whether the set of points $S= \{ s_1,\ldots,s_m \} $ is shared across images or independently defined for each image. I strongly recommend that the authors explicitly formalize the optimization problem for $\theta$, including the expectation over input images $I$. This clarification would help prevent potential misunderstandings of the proposed framework.

**Audience:**

Yes

**Audience Explanation:**

Vision Transformers (ViTs) have become a standard architecture in computer vision, and improving and analyzing their design remains an active and important research direction in the deep learning and computer vision communities. Therefore, I believe that the TMLR audience would be interested in the findings presented in this paper.

**Claims And Evidence:**

Yes

**Claims Explanation:**

As noted by all reviewers, the authors' claims are well supported by the experimental results, and the concerns raised during the review process have been adequately addressed through discussion.